# DDR: Exploiting Deep Degradation Response as Flexible Image Descriptor

**Juncheng Wu**[1,3]**, Zhangkai Ni**[1*]**, Hanli Wang**[1]**, Wenhan Yang**[2]**, Yuyin Zhou**[3]**, Shiqi Wang**[4]

[1] School of Computer Science and Technology, Tongji University, China
[2] Pengcheng Laboratory, China
[3] Department of Computer Science and Engineering, University of California, Santa Cruz, USA
[4] Department of Computer Science, City University of Hong Kong, Hong Kong

jwu418@ucsc.edu,    {zkni, hanliwang}@tongji.edu.cn
yangwh@pcl.ac.cn,    yzhou284@ucsc.edu,    shiqwang@cityu.edu.hk

## Abstract

Image deep features extracted by pre-trained networks are known to contain rich and informative representations. In this paper, we present Deep Degradation Response (DDR), a method to quantify changes in image deep features under varying degradation conditions. Specifically, our approach facilitates flexible and adaptive degradation, enabling the controlled synthesis of image degradation through text-driven prompts. Extensive evaluations demonstrate the versatility of DDR as an image descriptor, with strong correlations observed with key image attributes such as complexity, colorfulness, sharpness, and overall quality. Moreover, we demonstrate the efficacy of DDR across a spectrum of applications. It excels as a blind image quality assessment metric, outperforming existing methodologies across multiple datasets. Additionally, DDR serves as an effective unsupervised learning objective in image restoration tasks, yielding notable advancements in image deblurring and single-image super-resolution. Our code is available at: https://github.com/eezkni/DDR

## 1 Introduction

Deep features extracted by pre-trained neural networks are well-known for their capacity to encode rich and informative representations [1–4]. Extensive research efforts have aimed to quantify the information encoded within these deep features for use as image descriptors. For example, the distance between deep features has been employed as a metric for image quality assessment (IQA) in various studies [3, 5, 6]. Additionally, researchers have studied differences between images by comparing the distributions [2, 7] or frequency components [8, 9] of their deep features. Moreover, the statistical properties of deep features have been found to correlate with the style and texture of images in prior works [6, 10]. Recent research has also highlighted that the internal dissimilarity between deep features at different image scales can serve as a potent visual fingerprint [11].

This paper delves into an intriguing and unexplored property of image deep features: their response to degradation. Specifically, when subjecting images with diverse content and textures to various types of degradation, such as blur, noise, or JPEG compression, the deep features of these images exhibit varying degrees of change. This phenomenon is illustrated in Fig. 1, where Gaussian Blur is applied to images. The degrees of changes in feature space reflect the deep feature response to specific degradation. As shown in Tab. 1, a strong correlation exists between this response and the quality scores of blurred images. We validate that the response of deep features to degradation effectively

---

*Corresponding author: Zhangkai Ni (zkni@tongji.edu.cn)

38th Conference on Neural Information Processing Systems (NeurIPS 2024).

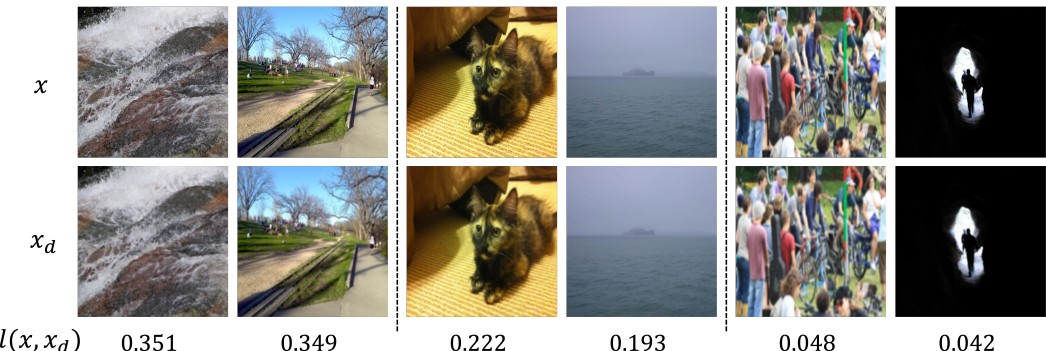

| | | | | | |
|---|---|---|---|---|---|
| $l(x, x_d)$ | 0.351 | 0.349 | 0.222 | 0.193 | 0.048 | 0.042 |

Figure 1: **Example of Degradation Response Variations.** We apply the same level of Gaussian Blur to different images from the LIVEitw [14] dataset. $x$ and $x_d$ denote the original and degraded images, respectively. $l(\cdot, \cdot)$ is the LPIPS metric [3] between $x$ and $x_d$, which measures the extent of changes in the feature space. The results demonstrate that images with different content and texture characteristics exhibit varying degrees of change.

| Metric | SRCC |
|---|---|
| ILNIQE [15] | 0.915 |
| WaDIQaM [16] | 0.938 |
| BIECON [17] | 0.956 |
| HOSA [18] | 0.954 |
| DBCNN [19] | 0.935 |
| HyperIQA [20] | 0.926 |
| DDR$_{blur}$ | **0.988** |

Table 1: **Comparison of SRCC for blur degradation on the LIVE [21] dataset.** DDR$_{blur}$ refers to the deep feature response to blur obtained by manually synthesizing degradation in the pixel domain. DDR$_{blur}$ demonstrates highest correlation with human opinion.

Figure 2: **Distribution of DDR on the LIVEitw [14] dataset.** "Low", "optimal", and "high" refer to different levels of handcrafted degradation applied in the pixel domain, while "adaptive" represents adaptively fusing text-driven degradation in the feature domain. The DDR with "optimal" and "adaptive" degradation achieve significantly better performance on the BIQA task.

captures different image characteristics by varying the type of degradation. Therefore, we propose the Deep Degradation Response (DDR), which quantifies the response of image deep features to specific degradation types, serving as a powerful and flexible image descriptor.

One straightforward approach to compute the DDR is to apply handcrafted degradation to the image, extract degraded features from the degraded image, and then calculate the distance between these degraded features and the features of the original image. However, as shown in Fig. 2, adjusting the level of degradation applied to the image significantly affects the distribution of DDR. Therefore, meticulous adjustment of the degradation level is crucial to achieve optimal performance in various downstream tasks. To address this challenge, we propose a text-driven degradation fusing strategy. Inspired by manifold data augmentation algorithms [12, 13], we adaptively fuse text features representing specific degradation onto the original image features, resulting directly in degraded features. By manipulating the text, we can effectively control the type of degradation fused to the image features. This approach allows us to flexibly assess the response of image features to various degradation types, thereby enhancing adaptability across different downstream tasks.

We evaluate the performance of our proposed DDR across multiple downstream tasks. Firstly, we assess its effectiveness as an image quality descriptor on the opinion-unaware blind image quality assessment (OU-BIQA) task, where DDR demonstrates superior performance compared to

existing OU-BIQA methods across various datasets. Secondly, we employ DDR as an unsupervised learning objective, training image restoration models specifically to maximize the DDR of the output image, which includes tasks such as image deblurring and real-world single-image super-resolution. Incorporating DDR as an external training objective consistently improves performance in both tasks, highlighting the strength of DDR as a flexible and powerful image descriptor.

## 2  Related Works

**Deep Feature Based Image Descriptors.** Image descriptors aim to quantify fundamental characteristics of images, such as texture [22], color [23, 24], complexity [25], and quality [26, 27]. With the informative representations in deep features extracted by pre-trained networks, various efforts have been made to develop image descriptors based on these features. Many existing descriptors regress the deep features of an image to a score, training the model by minimizing the loss between these predicted scores and the ground truth scores labeled by humans [25, 20, 28]. However, these methods are somewhat inflexible for two reasons: (1) they rely on human-labeled opinion scores, and (2) they are designed to evaluate fixed image characteristics. In this paper, we propose a flexible alternative by measuring the degradation response of deep features.

**Image Degradation Representation for Image Restoration.** Various deep learning-based image restoration methods leverage image degradation representation to enhance model performance [29–32]. For instance, some methods utilize contrastive-based [29] and codebook-based [30] techniques to encode various degradation types, enhancing the model's robustness to unknown forms of degradation. Moreover, other methods design degradation-aware modules to extract degradation representations from images and guide the removal of degradation [31, 32]. However, these methods depend on task-specific training. In contrast, the proposed DDR flexibly obtains representations for different types of degradation, making it an effective image descriptor for various image restoration tasks.

**Multimodal Vision Models.** Recent advancements in multimodal vision models, achieved through extensive training on paired image-text data [33–37], have significantly enhanced the capability of these models to understand and describe image textures using natural language [38]. Researchers have explored various methods to leverage this capability for modifying image texture attributes through language guidance. For instance, in language-guided image style transfer [39, 40], natural language descriptions are used to define the target texture style. Additionally, Moon et al. [12] introduced a manifold data augmentation technique that integrates language-guided attributes into image features. Building on these ideas, our work aims to adaptively fuse degradation information into image deep features using language guidance, thereby facilitating the measurement of our proposed DDR.

## 3  Deep Degradation Response as Flexible Image Descriptor

### 3.1  Deep Degradation Response

We define the Deep Degradation Response (DDR) as the measure of change in image deep features when specific types of degradation are introduced, which can be mathematically expressed as:

$$\text{DDR}_d\left(i\right) = \mathcal{M}\left(\mathcal{F}, \mathcal{F}_d\right), \tag{1}$$

where $d$ represents the type of degradation. $\mathcal{M}\left(\cdot, \cdot\right)$ denotes a disparity metric, such as $L_n$ distance or cosine distance. $\mathcal{F} = \Phi_v\left(i\right)$ represents the original image features extracted by a pre-trained network $\Phi_v(\cdot)$, while $\mathcal{F}_d$ denotes the degraded features.

The core of the proposed DDR lies in how to model $\mathcal{F}_d$. A naive approach to this involves synthesizing *degradation in the pixel domain*, *i.e.*, generating degraded images. As shown in Fig. 3 (a), a handcrafted degradation process is applied to the image, leading to the creation of a degraded image $i_d$. The extent of degradation is controlled by the parameter $\omega_d$. Then a pre-trained visual encoder $\Phi_v(\cdot)$ is utilized to extract the features of $i_d$, generating degraded features. Therefore, the pixel space degradation synthesizing process can be formulated as:

$$\mathcal{F}_d = \Phi_v\left(\text{D}\left(i, \omega_d\right)\right), \tag{2}$$

where $\text{D}(\cdot)$ denotes the handcrafted degradation synthesis process. However, as depicted in Fig. 2, varying levels of degradation significantly affect DDR. For downstream tasks, it is imperative

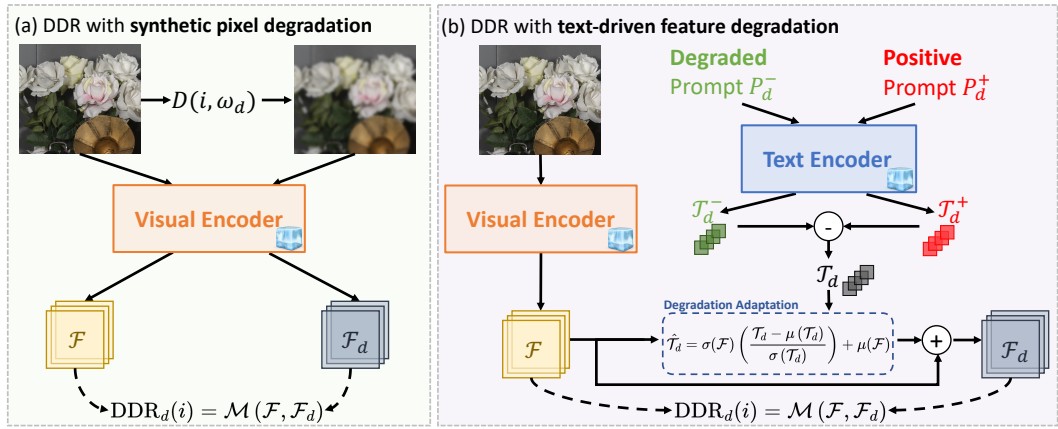

Figure 3: **The framework of our proposed DDR** with two different degradation fusing methods. (a) Synthesizing degradation with a handcrafted process *in the pixel domain*. (b) Fusing text-driven degradation *in the feature domain*.

to meticulously determine the optimal $\omega_d$ for different manual processes. This not only poses a substantial challenge but also diminishes the robustness of DDR as an image descriptor.

In this study, we propose a novel and efficient method for modeling $\mathcal{F}_d$ by synthesizing *degradation in the feature domain* using text-driven prompts. Specifically, to construct the degradation representation, we first design a pair of prompts: one describing an image with a specific type of degradation and the other describing the same image without degradation. These prompts are then separately encoded using the text encoder $\Phi_t(\cdot)$ of CLIP [37], yielding text-driven degradation representations $\mathcal{T}_d^-$ and $\mathcal{T}_d^+$, respectively. We obtain the degradation direction in the feature space by calculating the difference between these representations, as follows:

$$\mathcal{T}_d = \mathcal{T}_d^- - \mathcal{T}_d^+, \tag{3}$$

where $\mathcal{T}_d^- = \Phi_t(P_d^-)$ and $\mathcal{T}_d^+ = \Phi_t(P_d^+)$. $P_d^-$ and $P_d^+$ represent the degraded and clean prompts, respectively. However, due to the gap between text and image modality within the feature space [41, 42] of the CLIP model, we cannot effectively obtain the degraded image feature by directly fusing the features from different modalities. To address this challenge, we propose an adaptive degradation adaptation strategy by 'stylizing' the text-driven degradation representation using the image feature. Inspired by AdaIN [43], we propose to align the mean and variance of $\mathcal{T}_d$ to match those of the image feature, which can be formulated as follows:

$$\hat{\mathcal{T}}_d = \sigma(\mathcal{F}) \left( \frac{\mathcal{T}_d - \mu(\mathcal{T}_d)}{\sigma(\mathcal{T}_d)} \right) + \mu(\mathcal{F}), \tag{4}$$

where $\hat{\mathcal{T}}_d$ denotes the adapted degradation representation. Finally, we fuse the image feature with $\hat{\mathcal{T}}_d$, and the feature space text-driven degradation process can be represented as:

$$\mathcal{F}_d = \mathcal{F} + \hat{\mathcal{T}}_d. \tag{5}$$

Our proposed degradation fusion method allows us to measure DDR across various types of degradation simply by modifying the text prompt, eliminating the need for handcrafted design processes. Additionally, our adaptation strategy enables the application of text-driven degradation to image features without adjusting any hyper-parameters. As shown in Fig. 2, in the LIVEitw [14] dataset, DDR with text-driven feature degradation method achieves a distribution similar to DDR with carefully adjusted optimal degradation level, demonstrating the flexibility of our method.

### 3.2 DDR as a Flexible Image Descriptor

By modifying the degradation type, DDR can capture different characteristics in natural images. We demonstrate this by measuring DDR with different degradation types across all images in the

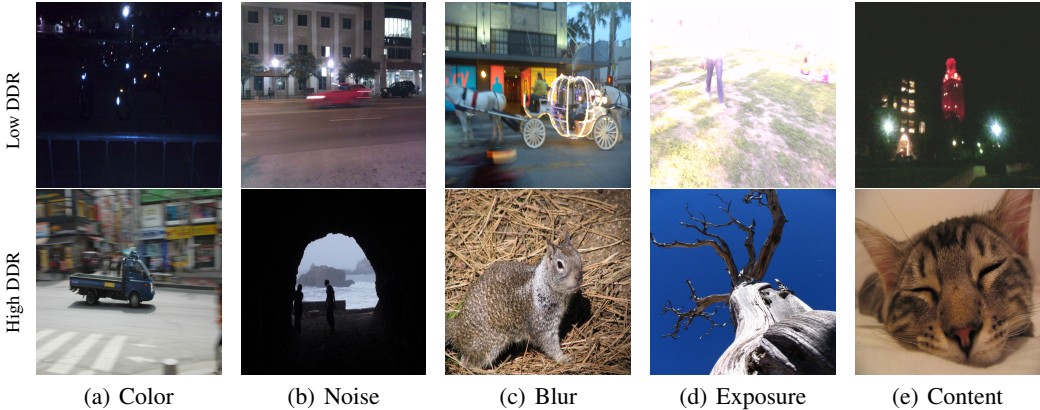

(a) Color    (b) Noise    (c) Blur    (d) Exposure    (e) Content

Figure 4: **Images with high and low DDR to different degradation types.** We measure DDR with five types of degradation by setting their corresponding prompt pair. We observe that image with lower DDR to a specific type of degradation is likely to obtain this degradation of a higher level.

| Degradation Type | Complexity [25] | Colorfulness [23] | Sharpness [44] | Quality |
|:---:|:---:|:---:|:---:|:---:|
| color | -0.223 | 0.757 | 0.715 | 0.790 |
| noise | -0.444 | 0.673 | 0.600 | 0.694 |
| blur | -0.206 | 0.612 | 0.732 | 0.756 |
| exposure | -0.435 | 0.684 | 0.699 | 0.770 |
| content | -0.357 | 0.612 | 0.561 | 0.642 |

Table 2: **SRCC between DDR and image characteristics.** With different types of degradation, DDR exhibits varying degrees of correlation with each image characteristic.

LIVEitw [14] dataset. Specifically, we set five pairs of prompts, representing five types of degradation, including color, noise, blur, exposure, and content. We employ a fixed prompt formatting for different types of degradation, as follows:

$$P_d^- = \text{A } \{d^-\} \text{ photo with low-quality}; \quad P_d^+ = \text{A } \{d^+\} \text{ photo with high-quality}. \tag{6}$$

For example, when the degradation type is blur, the $d^-$ and $d^+$ are set as 'blurry' and 'sharp' respectively. The images with high and low DDR for each type of degradation are shown in Fig. 4. We observe a negative correlation between the DDR and the level of degradation within the image. For example, as demonstrated in Fig. 4(e), an image with a high DDR to content degradation retains clear content, while the corresponding image with a low DDR exhibits unrecognizable content.

To further quantify the correlation between DDR and other image characteristics, we calculate the Spearman's Rank Correlation Coefficient (SRCC) between DDR and four types of image characteristics. Specifically, we measure the complexity [25], colorfulness [23], and sharpness [44] of images in the CSIQ [45] dataset. Additionally, we use the Mean Opinion Score (MOS) of each image as its quality score. The results are presented in Tab. 2. It is interesting to note that there is a negative correlation between the complexity of an image and DDR. This suggests that more complex images are capable of enduring more degradation with a smaller degree of change in deep features. Furthermore, the DDR to color and blur degradations show the highest correlation with colorfulness and sharpness respectively. Overall, with different degradation types, DDR tends to emphasize different image characteristics. Therefore, DDR shows promise as a versatile image descriptor for diverse downstream tasks through simple prompt adjustments, including IQA and image restoration.

### 3.3 DDR as a Blind Image Quality Assessment Metric

DDR can function as an image quality descriptor. As shown in Tab. 2, there is a positive correlation between the quality score and the DDR of the image. In cases where image quality is predominantly affected by a specific degradation type, an image with a high DDR to this degradation would likely

obtain a higher quality. For instance, in Fig. 4, when the degradation type is blur, comparing the image with a high DDR to the image with a low DDR, it is evident that the former exhibits a sharper content with less blur. However, real-world images often feature a mix of degradations. To evaluate image quality in such scenarios, we formulate a set of degradations denoted as $\mathcal{D}$ and compute the mean DDR for each degradation in $\mathcal{D}$. The blind image assessment metric based on DDR can thus be formulated as follows:

$$\text{Q}_{\text{DDR}}(i) = \frac{1}{|\mathcal{D}|} \sum_{d \in \mathcal{D}}^{d} \text{DDR}_d(i). \tag{7}$$

### 3.4 DDR as an Unsupervised Learning Objective

We can also utilize DDR as a learning objective in image restoration tasks, where the goal is to train a deep learning-based restoration model to predict a clean image from a degraded one. This is achieved by optimizing the restoration model to minimize the reconstruction loss function, which quantifies the difference between the pixel values of the model's output and the ground truth. In this work, we demonstrate that incorporating DDR as an external unsupervised learning objective can improve the optimization of the restoration models. Specifically, we measure the DDR of the model output and aim to simultaneously minimize the reconstruction loss while maximizing the DDR. The learning objective of the image restoration model is thus formulated as:

$$\min_{\theta} \left( \mathcal{L}_{rec}\left(R_\theta(i), i_{gt}\right) - \lambda_d \sum_{d \in \mathcal{D}}^{d} \text{DDR}_d\left(R_\theta(i)\right) \right), \tag{8}$$

where $R_\theta(\cdot)$ is a restoration model parameterised by $\theta$, and $\mathcal{L}_{rec}\left(\cdot, \cdot\right)$ denotes the reconstruction loss, $\lambda_d$ is the weight of DDR in learning objective. Similarly, by adjusting the degradation prompt and combining different types of degradation, we can tailor the approach to various restoration tasks.

## 4 Experiments

### 4.1 Experiment Setting

To demonstrate the versatility and efficacy of DDR as an image descriptor, we conduct comprehensive experiments covering (1) opinion-unaware blind image quality assessment (OU-BIQA), which does not require training model with human-labeled Mean Opinion Score (MOS) values, and (2) image restoration tasks, including image deblurring and real-world image super-resolution.

**Implementation Details.** For different tasks, we tailor the degradation set $\mathcal{D}$ in Eq. 7 and Eq. 8 to focus on distinct image attributes. Specifically, in BIQA, we define $\mathcal{D} = \{$**color**, **noise**, **blur**, **exposure**$\}$. Meanwhile, for image restoration tasks, we set $\mathcal{D} = \{$**color**, **content**, **blur**$\}$. In all experiments related to image restoration, we empirically set the weight of DDR in the learning objective in Eq. 8 as $\lambda_d = 2.0$. We use the CLIP [37] ViT-B/32 model as the image feature extractor, and employ the cosine distance to quantify the disparity between original and degraded image features, which can be defined as follows:

$$\mathcal{M}_{cos}(x, y) = 1 - \frac{x \cdot y}{\|x\|\|y\|}, \tag{9}$$

**Baseline Datasets.** To evaluate the effectiveness of the proposed DDR as an image quality descriptor, we conduct extensive experiments on eight public IQA datasets, including CSIQ [45], TID2013 [46], KADID [47], KonIQ [48], LIVE in-the-wild [14], LIVE [21], CID2013 [49], and SPAQ [50], which encompass both synthetic and real-world degradation scenarios. For image deblurring, we train and test the model using the GoPro dataset [51] and RealBlur dataset [52], respectively. The GoPro dataset [51] consists of synthetic blurred images, while RealBlur [52] contains images with real-world motion blur. For SISR, we combine two real-world datasets together for training and testing, including the RealSR [53] and City100 [54] datasets.

**Baseline Methods.** For the OU-BIQA task, we compare DDR with representative and state-of-the-art opinion-unaware BIQA (OU-BIQA) methods, which do not require training with human-labeled MOS. The compared methods include NIQE [26], QAC [55], PIQE [56], LPSI [57], ILNIQE [15], diqIQ [58], SNP-NIQE [59], NPQI [60], and ContentSep [61]. Among all compared methods, DDR

| Datasets | NIQE | QAC | PIQE | LPSI | ILNIQE | dipIQ | SNP-NIQE | NPQI | ContentSep | Ours |
|---|---|---|---|---|---|---|---|---|---|---|
| CSIQ | 0.6191 | 0.4804 | 0.5120 | 0.5218 | 0.8045 | 0.5191 | 0.6090 | 0.6341 | 0.5871 | **0.8289** |
| LIVE | 0.9062 | 0.8683 | 0.8398 | 0.8181 | 0.8975 | **0.9378** | 0.9073 | 0.9108 | 0.7478 | 0.8793 |
| TID2013 | 0.3106 | 0.3719 | 0.3636 | 0.3949 | 0.4938 | 0.4377 | 0.3329 | 0.2804 | 0.2530 | **0.5844** |
| KADID | 0.3779 | 0.2394 | 0.2372 | 0.1478 | 0.5406 | 0.2977 | 0.3719 | 0.3909 | 0.5060 | **0.5968** |
| KonIQ | 0.5300 | 0.3397 | 0.2452 | 0.2239 | 0.5057 | 0.2375 | 0.6284 | 0.6132 | 0.6401 | **0.6455** |
| LIVEitw | 0.4495 | 0.2258 | 0.2325 | 0.0832 | 0.4393 | 0.2089 | 0.4654 | 0.4752 | 0.5060 | **0.6613** |
| CID2013 | 0.6589 | 0.0299 | 0.0448 | 0.3229 | 0.3062 | 0.3776 | 0.7159 | 0.7698 | 0.6116 | **0.8009** |
| SPAQ | 0.3105 | 0.4397 | 0.2317 | 0.0001 | 0.6959 | 0.2189 | 0.5402 | 0.5999 | 0.7084 | **0.7249** |

Table 3: **Quantitative result of OU-BIQA.** Performance comparisons of different OU-BIQA models on eight public datasets using SRCC. The top performer on each dataset is marked in **bold**.

is the only *zero-shot* method that does not require any training. For image restoration, we compare our proposed method, as illustrated in Eq. 8, with a combination of reconstruction loss and feature domain loss $\mathcal{L}_f(\cdot, \cdot)$, which quantifies the distance between deep features extracted from images. Generally, the reconstruction loss is combined with feature domain losses to enhance the overall quality of the restored image, forming the learning objective of the restoration model $R_\theta(\cdot)$ as follows:

$$\min_\theta \left( \mathcal{L}_{rec}\left(R_\theta(i), i_{gt}\right) + \lambda_f \mathcal{L}_f\left(R_\theta(i), i_{gt}\right) \right),\tag{10}$$

where $\lambda_f$ is the weighting factor for $\mathcal{L}_f(\cdot, \cdot)$. In all experiments on image restoration, we utilize PSNR loss as the reconstruction loss. We consider four types of representative feature domain losses for comparison, including LPIPS [3], CTX [2], PDL [7], and FDL [8]. To ensure a fair comparison, we set $\lambda_f = 0.1$ for FDL [8] and $\lambda_f = 1.0$ for the other feature domain losses, ensuring that the magnitudes of the different feature domain losses are in a similar range.

Moreover, to fully assess the robustness of our proposed DDR across vaious architectural models, we conduct all image restoration experiments using two representative image restoration models: NAFNet [62] and Restormer [63]. NAFNet [62] is a convolutional neural network (CNN)-based model, while Restormer [63] is a Transformer [64]-based model. These models have demonstrated impressive performance in their respective tasks and are widely recognized as representative models in recent years. We empirically train the model at a resolution of $128 \times 128$. For the learning rate, we adhere to the official settings for NAFNet and Restormer. Specifically, the initial learning rate for NAFNet is set to $1e-3$, and for Restormer, it is set to $3e-4$. We also adopted a cosine annealing strategy for both models.

### 4.2 Opinion-Unaware Blind Image Quality Assessment

Tab. 3 presents the results across all datasets. Our proposed DDR consistently outperforms all competing methods on datasets with both synthetic [45–47] and in-the-wild [14, 49, 48, 50] degradation, underscoring its robustness across diverse degradation types. Especially its substantial improvement in SRCC on the LIVE in-the-wild dataset, rising from 0.5060 to 0.6613, showcasing the effectiveness of DDR as an image quality descriptor for images with real-world degradation. Furthermore, comparing the SRCC performance in Tab. 3 and Tab. 2, it is obvious that on the CSIQ dataset [45], DDRs that integrate multiple types of degradation perform significantly better than DDRs that only focus on a single type of degradation. This underscores the superiority of DDR as a more comprehensive image quality descriptor simply by integrating multiple types of degradation.

### 4.3 Image Motion Deblurring

The objective of image deblurring is to restore a high-quality image with clear details. The quantitative analysis in Tab. 4 illustrates that our proposed DDR surpasses all compared loss functions across datasets with both synthetic and real-world blur. Compared to optimizing solely the PSNR loss, our approach achieves a notable enhancement in PSNR, with an increase of at least 0.16 dB across all models and datasets. These results suggest that maximizing the DDR of the predicted image results in higher fidelity and reduced degradation. This is further evident in the qualitative results shown in Fig. 5, where the PSNR loss alone produces blurry textures, and combining PSNR with feature

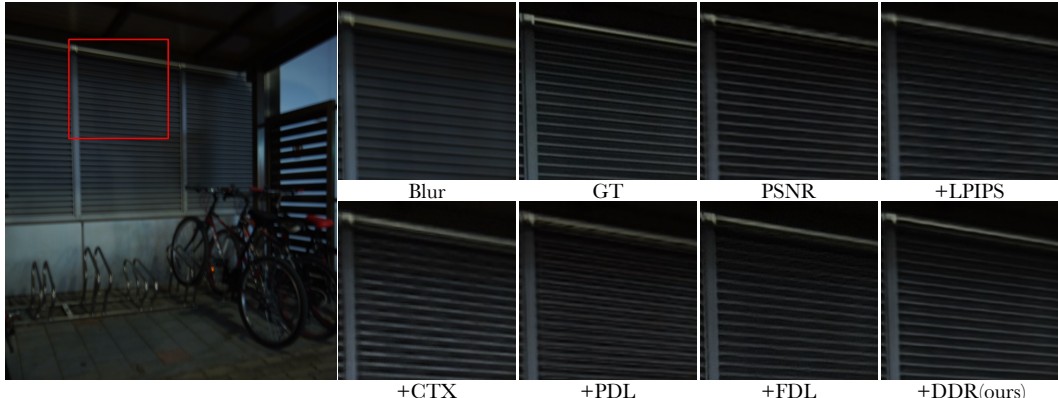

Figure 5: **Qualitative result on RealBlur [52] dataset.** The training of model is supervised by (1) reconstruction loss (PSNR) (2) reconstruction loss combined with feature domain loss, and (3) reconstruction loss combined with DDR. The red area is cropped from different results and enlarged for visual convenient. Appending DDR as an external self-supervised learning objective leads to result with more natural texture and less artifacts.

| model | loss | GoPro [51] | | RealBlur [52] | |
| --- | --- | --- | --- | --- | --- |
| | | PSNR | SSIM | PSNR | SSIM |
| NAFNet | PSNR | 33.1717 | 0.9482 | 30.6373 | 0.9038 |
| | PSNR + LPIPS [3] | 33.1660 | 0.9481 | 30.7245 | 0.9044 |
| | PSNR + CTX [2] | 32.7879 | 0.9436 | 30.4394 | 0.8985 |
| | PSNR + PDL [7] | 32.9417 | 0.9463 | 30.6270 | 0.9039 |
| | PSNR + FDL [8] | 32.8321 | 0.9420 | 30.1743 | 0.8864 |
| | PSNR + DDR(ours) | **33.3427** | **0.9500** | **30.7982** | **0.9049** |
| Restormer | PSNR | 33.3398 | 0.9494 | 31.9816 | 0.9098 |
| | PSNR + LPIPS [3] | 33.3717 | 0.9495 | 31.9639 | 0.9099 |
| | PSNR + CTX [2] | 33.2834 | 0.9483 | 31.9893 | 0.9101 |
| | PSNR + PDL [7] | 33.2905 | 0.9487 | 31.9900 | 0.9106 |
| | PSNR + FDL [8] | 33.3560 | 0.9489 | 31.7673 | 0.9034 |
| | PSNR + DDR(ours) | **33.4946** | **0.9513** | **32.1759** | **0.9121** |

Table 4: **Quantitative result of image motion deblurring.** Experiment is conducted on datasets with synthetic [51] and real-world [52] blur respectively. The best results are marked in **bold**. Combining proposed DDR with reconstruction loss leads to result with less degradation and higher fidelity. Our proposed method demonstrates the robustness to model architecture and dataset.

domain losses introduces noticeable artifacts. In contrast, incorporating DDR substantially reduces artifacts, yielding predicted images with sharper and more natural textures.

## 4.4 Single Image Super Resolution

SISR is a task aimed at enhancing the resolution of a low-resolution image to match or surpass the quality of a high-resolution counterpart. In our study, we evaluate our proposed DDR method against state-of-the-art loss functions. Tab. 5 showcases the quantitative results on a real-world dataset by two representative models (NAFNet and Restormer). Our findings reveal that our method outperforms all competing methods in terms of PSNR. Particularly noteworthy is the improvement achieved with NAFNet, where the incorporation of DDR alongside the reconstruction loss elevates the PSNR from 27.08 to 27.31. Additionally, as depicted in Fig. 6, our method yields visual results with finer texture compared to those optimized solely for PSNR or combined with LPIPS.

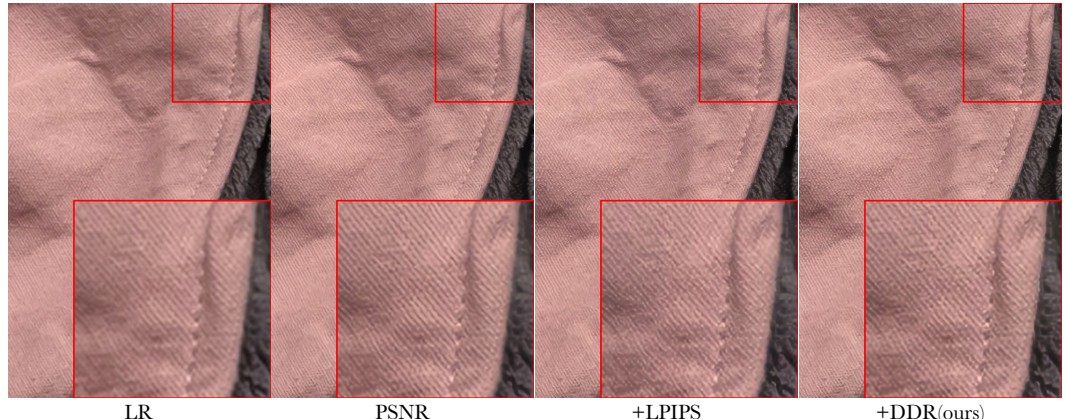

| | LR | PSNR | +LPIPS | +DDR(ours) |

Figure 6: **Qualitative result on real-world SISR dataset [53, 54].** DDR leads to results with sharper texture.

| loss | NAFNet [62] | | Restormer [63] | |
|------|-------------|------|----------------|------|
| | PSNR | SSIM | PSNR | SSIM |
| PSNR | 27.0856 | 0.8917 | 28.1491 | 0.8986 |
| PSNR + LPIPS [3] | 27.2835 | **0.8938** | 28.1221 | 0.8985 |
| PSNR + CTX [2] | 27.0985 | 0.8867 | 28.0933 | 0.8964 |
| PSNR + PDL [7] | 26.9467 | 0.8907 | 28.1413 | 0.8985 |
| PSNR + FDL [8] | 27.0263 | 0.8809 | 28.0746 | 0.8966 |
| PSNR + DDR(ours) | **27.3121** | 0.8923 | **28.1668** | **0.8990** |

Table 5: **Quantitative result on real-world SISR dataset [53, 54].**

## 4.5 Ablation Study

For image deblurring, we conduct a series of ablation experiments on the NAFNet using the GoPro dataset, with all results detailed in Table 6. Firstly, we adjusted $\mathcal{D}$ to evaluate the effect of the degradation set defined in Eq. 8. Notably, a decrease in performance is observed when any type of degradation is removed, suggesting that combining multiple types of degradation results in a more comprehensive image description. Secondly, we investigate the effect of $\lambda_d$ in Eq. 8. Minor fluctuations in performance are observed when adjusting $\lambda_d$ to 1.0 and 3.0, indicating the robustness of our method to this hyper-parameter. Next, we explore the effect of the visual feature extractor. Increasing the scale of $\Phi_v$ in DDR does not lead to improved performance, suggesting that a larger visual model may not necessarily enhance the ability to understand low-level texture. Finally, we examine the impact of the adaptation strategy in Eq. 4. A significant drop in performance is observed when the adaptation is removed, highlighting the critical role of this strategy in DDR calculation.

For opinion-unaware blind image quality assessment task, we conduct ablation experiment on four datasets. As demonstrated in Tab. 7, comparing with measuring DDR to single type of degradation, combining mutiple degradation types consistently leads to significant performance improvement. Furthermore, we can observe a performance boost by utilizing degradation adaptation strategy, improving SRCC from 0.6074 to 0.8289 on CSIQ dataset.

## 5 Limitations and Discussion

In this section, we discuss the limitations of DDR and provide potential solutions to address them.

**The ability of the visual feature extractor to understand low-level degradation.** We currently employ the CLIP model's visual feature extractor to facilitate text-driven degradation fusion. These feature extractors may incline to focus on high-level information such as image content, while their ability to understand low-level degradation may be limited. This could impact the measurement of the degradation response. In future work, we plan to fine-tune the feature extractor on tasks such

| $\mathcal{D}$ | $\lambda_d$ | Backbone | Adaptation | PSNR | SSIM |
|---|---|---|---|---|---|
| {**blur**, **content**} | 2.0 | ViT-B/32 | ✓ | 33.2080 | 0.9482 |
| {**color**, **content**} | 2.0 | ViT-B/32 | ✓ | 33.2077 | 0.9483 |
| {**color**, **blur**} | 2.0 | ViT-B/32 | ✓ | 33.2912 | 0.9492 |
| {**color**, **blur**, **content**} | 1.0 | ViT-B/32 | ✓ | 33.3031 | 0.9494 |
| {**color**, **blur**, **content**} | 3.0 | ViT-B/32 | ✓ | 33.3419 | 0.9495 |
| {**color**, **blur**, **content**} | 2.0 | ViT-B/16 | ✓ | 33.1194 | 0.9467 |
| {**color**, **blur**, **content**} | 2.0 | ViT-L/14 | ✓ | 33.0426 | 0.9465 |
| {**color**, **blur**, **content**} | 2.0 | RN50x16 | ✓ | 33.2709 | 0.9490 |
| {**color**, **blur**, **content**} | 2.0 | RN50x64 | ✓ | 33.1379 | 0.9472 |
| {**color**, **blur**, **content**} | 2.0 | ViT-B/32 | ✗ | 33.0864 | 0.9469 |
| {**color**, **blur**, **content**} | 2.0 | ViT-B/32 | ✓ | **33.3427** | **0.9500** |

Table 6: **Ablation results on GoPro [51] dataset and NAFNet [62].**

| $\mathcal{D}$ | Backbone | Adaptation | CSIQ | TID2013 | LIVEitw | KonIQ |
|---|---|---|---|---|---|---|
| {**color**} | ViT-B/32 | ✓ | 0.7896 | 0.4872 | 0.5178 | 0.5745 |
| {**blur**} | ViT-B/32 | ✓ | 0.7555 | 0.5061 | 0.6282 | 0.6028 |
| {**noise**} | ViT-B/32 | ✓ | 0.6937 | 0.5022 | 0.4954 | 0.5804 |
| {**exposure**} | ViT-B/32 | ✓ | 0.7695 | 0.5440 | 0.6312 | 0.5758 |
| {**blur**, **noise**, **exposure**} | ViT-B/32 | ✓ | 0.8029 | 0.5841 | 0.6660 | 0.6390 |
| {**color**, **noise**, **exposure**} | ViT-B/32 | ✓ | 0.8100 | 0.5917 | 0.6352 | 0.6379 |
| {**color**, **blur**, **exposure**} | ViT-B/32 | ✓ | 0.8054 | 0.5446 | 0.6548 | 0.6229 |
| {**color**, **blur**, **noise**} | ViT-B/32 | ✓ | 0.8157 | 0.5824 | 0.6399 | 0.6526 |
| {**color**, **blur**, **noise**, **exposure**} | ViT-B/32 | ✗ | 0.6074 | 0.4121 | 0.3437 | 0.4310 |
| {**color**, **blur**, **noise**, **exposure**} | ViT-B/16 | ✓ | 0.7841 | **0.6251** | **0.6888** | **0.6635** |
| {**color**, **blur**, **noise**, **exposure**} | ViT-B/32 | ✓ | **0.8289** | 0.5844 | 0.6613 | 0.6455 |
| {**color**, **blur**, **noise**, **exposure**} | RN50x16 | ✓ | 0.6500 | 0.5464 | 0.6607 | 0.6125 |
| {**color**, **blur**, **noise**, **exposure**} | RN50x64 | ✓ | 0.6162 | 0.5607 | 0.6703 | 0.6109 |

Table 7: **Ablation results on opinion unaware blind image quality assessment.**

as degradation classification or description, to enable it to extract more fine-grained degradation features.

**The selection of degradation prompts for different downstream tasks.** The suitable degradation prompts may vary for different downstream tasks. In future work, we hope to append learnable tokens in the degradation prompts, and fine-tune these tokens to better adapt our method to different tasks. Specifically, we can utilize the strategy such as adversarial training, training DDR as a discriminator. This is an interesting and promising direction for our further investigation.

## 6 Conclusions

This paper introduces a flexible and powerful image descriptor, which measures the response of image deep features to degradation. We propose a text-driven approach to adaptively fuse degradation into image features. Experimental results demonstrate that DDR achieves state-of-the-art performance in blind image quality assessment task, and optimizing DDR results in images with reduced distortion and improved overall quality in image restoration tasks. We believe that DDR can facilitate a better understanding and application of image deep features.

**Acknowledgments** This work was supported in part by the National Natural Science Foundation of China under Grant 62201387, in part by the Shanghai Pujiang Program under Grant 22PJ1413300, and in part by the Fundamental Research Funds for the Central Universities.

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

# A Appendix

## A.1 Detailed Experiment Settings

### A.1.1 Evaluation Metrics.

For BIQA, we select SRCC as evaluation metric to measure the correlation between predicted quality score and human opinion. For image restoration, we choose PSNR and Structural Similarity (SSIM) [65] as evaluation metrics to measure the fidelity and structure similarity of restored images, respectively.

### A.1.2 Training Settings

For all experiment on image restoration, we employ AdamW [66] optimizer and set the batch size as 4. We train NAFNet [62] and Restormer [63] for 200,000 and 300,000 steps respectively, following their corresponding official settings. All experiments are conducted using one NVIDIA RTX 4090. Each setting in image restoration costs about 20 to 30 hours for training.

### A.1.3 Explanation of Fig. 2

We measure the amount of images with different Degradation Response (DDR) values to represent the distribution of DDR. Specifically, we divided the range of DDR into multiple intervals. For each point on the curve, we measured the number of images whose DDR values fall within the corresponding interval. The horizontal axis in Fig. 2 represents the numerical values of DDR, while the vertical axis represents the number of images. By adjusting the levels of handcrafted degradation, DDR demonstrates varying performance on the Opinion-Unaware Blind Image Quality Assessment (OU-BIQA) task. We conducted experiments across a range of degradation levels, selecting the level with the best performance on OU-BIQA as "optimal". "Low" and "high" represent the lowest and highest degradation levels, respectively. When the degradation level is too low, there is only a subtle difference between the $\mathcal{F}$ and $\mathcal{F}_d$ for all images. In contrast, when the degradation level is too high, most images demonstrate an overly strong response. Using our text-driven "adaptive" strategy, DDR demonstrates a similar value distribution and performance to the manually set "optimal" degradation level. This result shows the effectiveness and flexibility of the proposed method.

### A.1.4 Degraded and Positive Prompt Pairs

Full degraded and positive prompt pairs are shown in Tab. 8. We set each prompt following the format in Eq. 6.

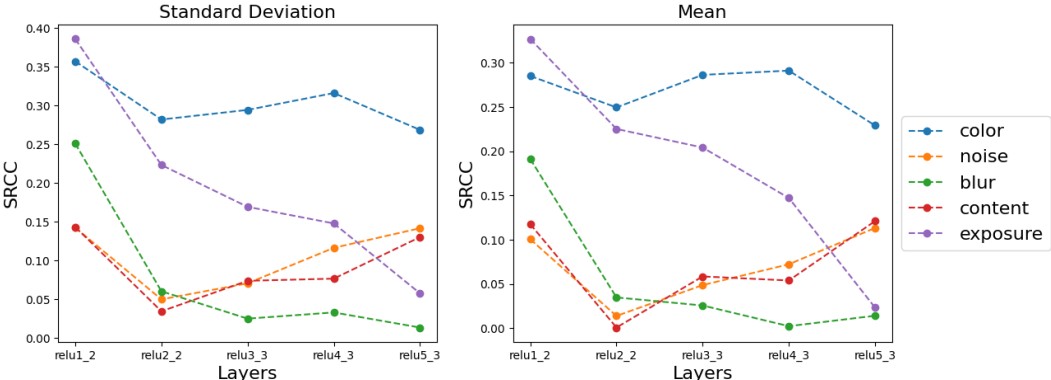

Figure 7: **SRCC between DDR and statistics of deep features.** Deep features are extracted from different layers of pre-trained VGG [67] network.

## A.2 Additional Experiment Results

Here we present additional experiment results. Firstly, statistics of deep features is known to be correlated with multiple image characteristics such as texture and style [6, 10]. Therefore, it is

| Degradation Type | Degraded Prompt |
|---|---|
| color | A **unnatural color** photo with low-quality. |
| noise | A **noise degraded** photo with low-quality. |
| blur | A **blurry** photo with low-quality. |
| exposure | A **unnatural exposure** photo with low-quality. |
| content | A **bad content** photo with low-quality. |

| Degradation Type | Positive Prompt |
|---|---|
| color | A **real color** photo with high-quality. |
| noise | A **clean** photo with high-quality. |
| blur | A **sharp** photo with high-quality. |
| exposure | A **natural exposure** photo with high-quality. |
| content | A **clear content** photo with high-quality. |

Table 8: **Degraded and Positive Prompts pairs in our experiment**.

interesting to investigate the correlation between DDR and deep feature statistics. Specifically, we extract features from five layers ($Relu\_1\_1$, $Relu\_2\_1$, $Relu\_3\_1$, $Relu\_4\_1$, and $Relu\_5\_1$) of pretrained VGG [67] network, and measure the mean and standard deviation of extracted features. Then, we utilize SRCC to quantify the correlation between DDR and these statistics. The results are shown in Fig. 7. We can observe that DDR to color degradation demonstrates similar correlation to statistics of feature from every layers. While for exposure and blur degradation, DDR shows significant higher correlation to mean and standard deviation of feature from $Relu\_1\_1$ than subsequent layers. In contrast, for noise and content degradation, DDR shows higher correlation for $Relu\_1\_1$ and $Relu\_5\_1$.

Secondly, we present additional qualitative results for image restoration tasks. Fig. 9 and Fig. 8 show the results in image deblurring on the realBlur [52] dataset and GoPro [51] dataset respectively. Moreover, the qualitative results in SISR on real-world dataset [53, 54] are demonstrated in Fig. 10.

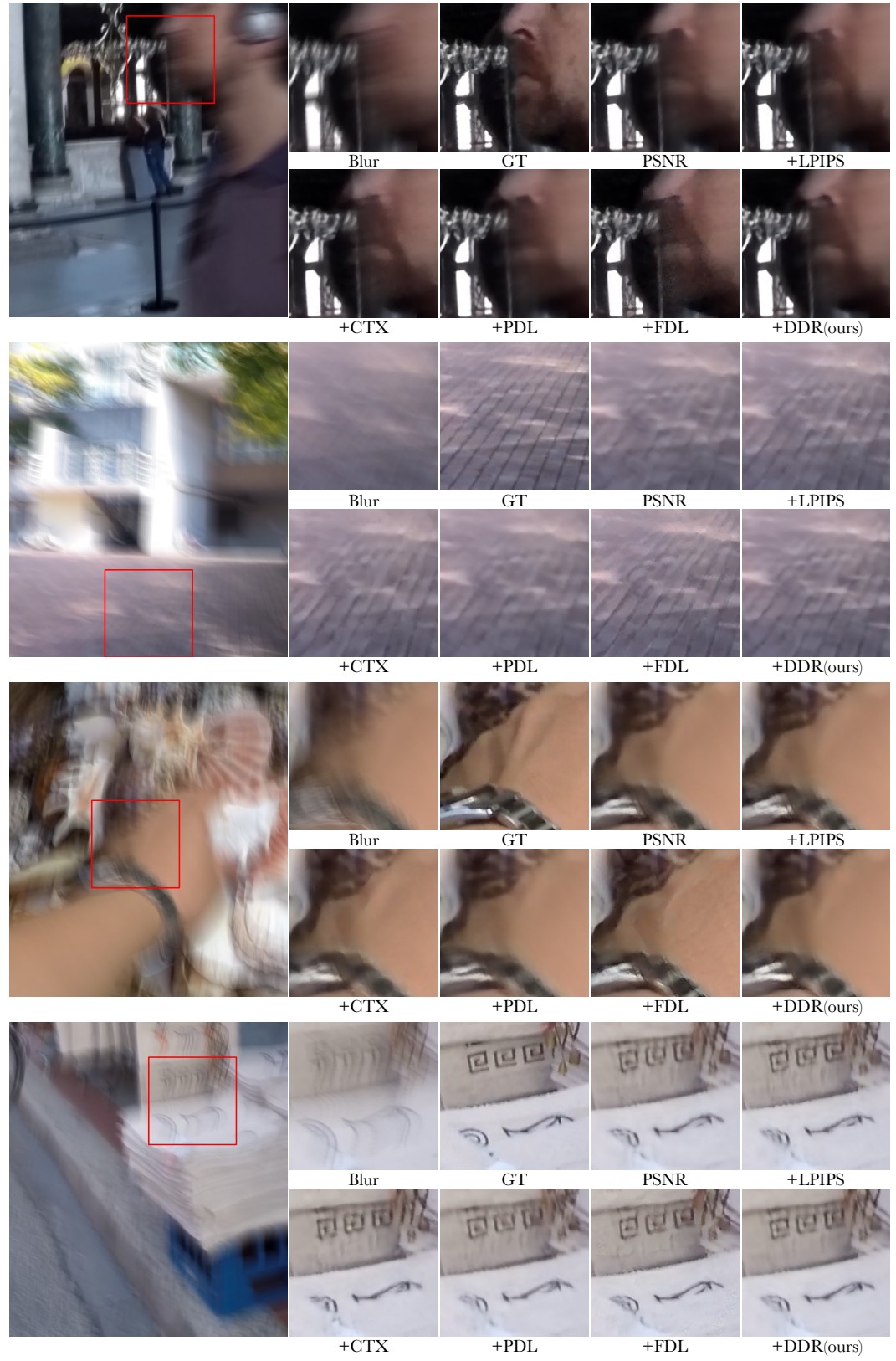

Figure 8: **Qualitative result of image deblurring using the NAFNet [62] trained with GoPro [51] dataset.** The red area is cropped from different results and enlarged for visual convenient.

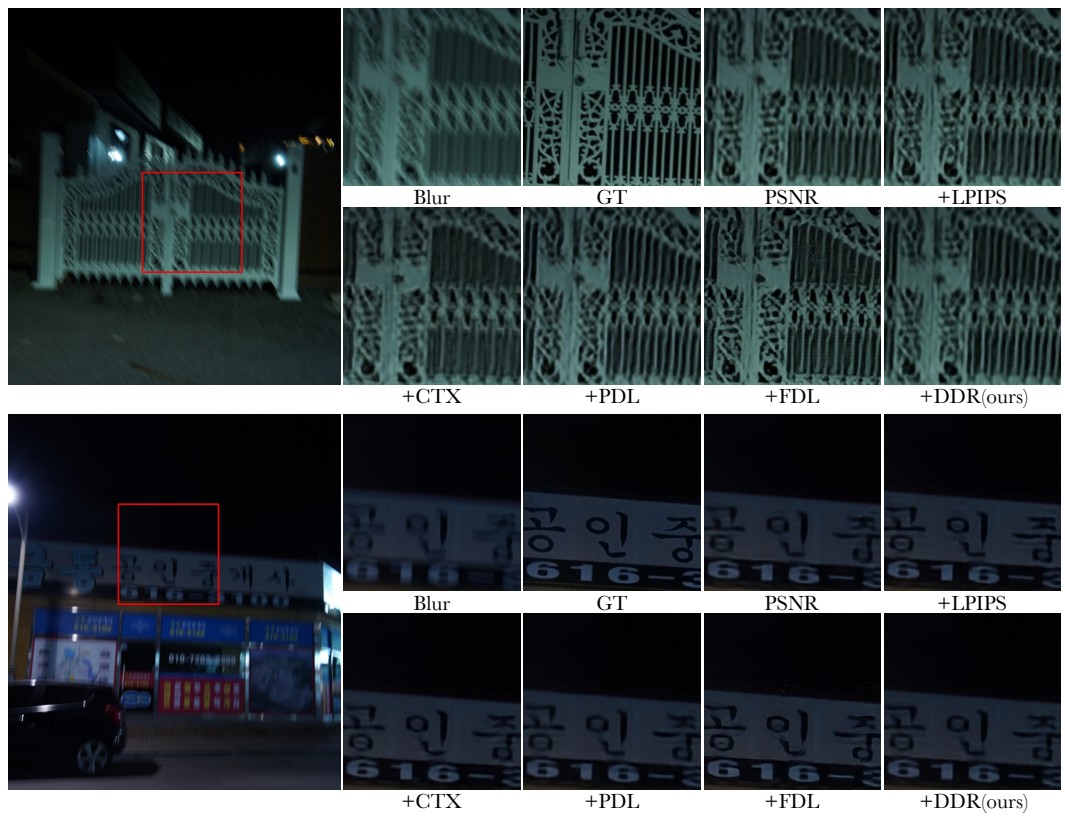

Figure 9: **Qualitative result of image deblurring using the NAFNet [62] trained with realBlur [52] dataset.** The red area is cropped from different results and enlarged for visual convenient.

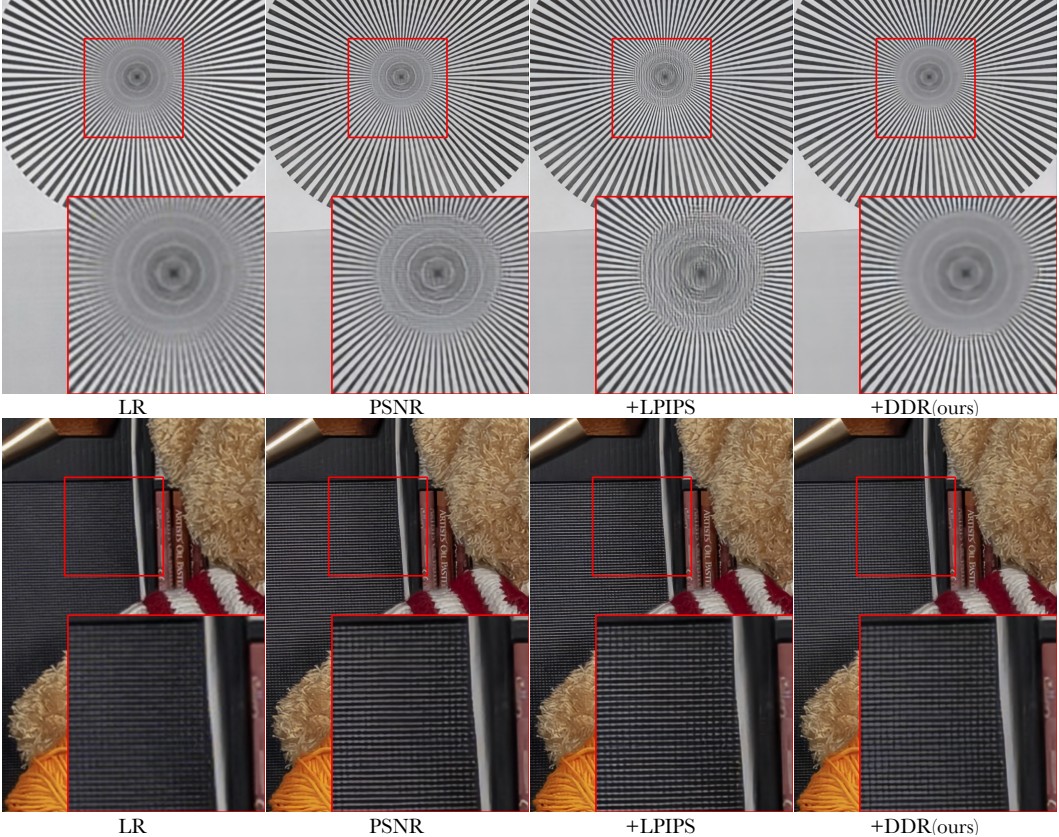

Figure 10: **Qualitative result of SISR using the NAFNet [62] trained with real-world SISR [53, 54] dataset.** The red area is cropped from different results and enlarged for visual convenient.

