# OpenReview forum: "DDR: Exploiting Deep Degradation Response as Flexible Image Descriptor"
_NeurIPS.cc/2024/Conference — NeurIPS 2024 poster_

### Official Review · Reviewer_tuJD · 2024-07-11

**Soundness:** 3
**Presentation:** 3
**Contribution:** 2
**Rating:** 7
**Confidence:** 4

**Summary:**

This paper proposed a low-level visual descriptor with text embeddings and explored its application on many low-level vision tasks.

**Strengths:**

1. The experiment is sufficient and detailed.
2. The paper is well-written.

**Weaknesses:**

**Weaknesses of the methods.**

1. This paper uses a text-based model to describe the low-level visual information of the images, which seems to be similar to Q-Bench [1]. I suggest that the author compare it with it and emphasize the differences between the method proposed and Q-Bench.
2. Simple super-resolution networks can identify different low-level visual information [2,3], such as degradation types. The author should compare with this method. Also, the description of lines 69 and 70 could be appropriately modified based on this.
3. The network settings for single-image super-resolution (SISR) are somewhat unreasonable. NAFNet and Restormer have a lot of downsampling and are not suitable as base models for SISR experiments. I suggest to use RCAN, SwinIR, and HAT.

[1] Wu H, Zhang Z, Zhang E, et al. Q-Bench: A Benchmark for General-Purpose Foundation Models on Low-level Vision.

[2] Liu Y, Liu A, Gu J, et al. Discovering distinctive" semantics" in super-resolution networks.

[3] Liu Y, He J, Gu J, et al. Degae: A new pretraining paradigm for low-level vision.

**Questions:**

1. The order of the figures and tables in this paper seems a little confusing and should be more carefully formatted. For example, the experimental results of SISR are not introduced until the end of page 8, but the result images are shown at the beginning of page 8.
2. The experimental settings are not fully given, such as the learning rate and training resolution, which may lead to unfair comparison. For example, SISR models are often trained on 48 $\times$ 48 resolution, are the NAFNet and Restormer also trained on 48 resolution for SISR tasks in this paper?

**Limitations:**

Please see the weaknesses and questions.

---

> ### Author Rebuttal · Authors · 2024-08-07
>
> We thank the reviewer for the thoughtful response to our paper. We address specific points below:
>
> Q1. **Comparison with Q-bench:**
>
> Thank you for the suggestion. There are key differences in **target** and **methodology** between the proposed DDR and Q-bench [r1] (in this response letter):
>
> - **Target:** Our goal is to measure the deep degradation response in the feature space as an image descriptor, while Q-bench aims to evaluate the low-level capabilities of existing Multi-modal Large Language Models (MLLMs).
>
> - **Methodology:** Our proposed method uses CLIP to encode degradation prompts and calculates the response of image deep features to these text-driven degradation representations. In contrast, Q-bench directly asks MLLMs to generate text describing low-level information within the images.
>
> Based on your suggestion, we will emphasize these differences in our revision.
>
> [r1] Wu H, Zhang Z, Zhang E, et al. Q-Bench: A Benchmark for General-Purpose Foundation Models on Low-level Vision.
>
> Q2. **Comparison with mentioned papers:**
>
> Thank you for the helpful suggestion. Reference [r2] (in this response letter) reveals that a pre-trained Super-Resolution (SR) network is capable of extracting degradation-related representations. Similarly, [r3] (in this response letter) uses a pre-trained SR network to extract degradation representations from images with various types of degradation. In comparison, our proposed method uses the text encoder of the CLIP model to encode degradation prompts as degradation representations. We will cite and discuss [r2] and [r3] in our revision and revise lines 69 and 70 accordingly.
>
> [r2] Liu Y, Liu A, Gu J, et al. Discovering distinctive ‘semantic’ in super-resolution networks.
>
> [r3] Liu Y, He J, Gu J, et al. Degae: A new pretraining paradigm for low-level vision.
>
> Q3. **Network Setting for SISR:**
>
> Thank you for the helpful suggestion. This work focuses on single-image super-resolution (SISR), where low-resolution (LR) and high-resolution (HR) images are with the same resolution, whereas HAT and RCAN focus on classical SISR that expands the resolution of the input image. Based on your suggestion, we conducted experiments using SwinIR. The results are shown in Table 3 (in separate PDF file), where we observe that the proposed DDR also achieves the best performance.
>
> Q4. **Formatting of Figures and Table:**
>
> Thank you for the helpful suggestion. We will carefully reformat the results Tables and Figures.
>
> Q5. **More experiment details:**
>
> According to your suggestion, we provide more experimental details here and will also explain them in the revision. Unlike classical single-image super-resolution (SISR), we conduct experiments on real-world image super-resolution datasets that low-resolution (LR) and high-resolution (HR) images are with the same resolution. Therefore, we empirically train the model at a resolution of 128 × 128. For the learning rate, we adhere to the official settings for NAFNet and Restormer. Specifically, the initial learning rate for NAFNet is set to 1e-3, and for Restormer, it is set to 3e-4. We also adopted a cosine annealing strategy for both models.

---

> > ### Comment · Reviewer_tuJD · 2024-08-12
> >
> > Thanks for authors' response on the comments. I still have some concerns about DDR.
> >
> > **Comparison with Q-bench.** Could we think that Q-Bench explicitly aligns the two modalities of degradation and text, while DDR aligns these two modalities in latent space?

---

> > > ### Author Response · Authors · 2024-08-12
> > >
> > > We appreciate for your thoughtful comments and insights.
> > >
> > > Regarding your concern towards the comparison with Q-bench, we acknowledge that both Q-bench and the proposed DDR approach necessitate alignment between modalities of degradation and text. Specifically, Q-bench explicitly instructs MLLMs to align these modalities, while DDR requires alignment in latent space to facilitate the generation of degradation representations. Your insights have provided a valuable perspective on this issue.

---

> > > > ### Comment · Reviewer_tuJD · 2024-08-12
> > > >
> > > > This difference shows that this paper contributed to the IQA field. My concerns have been addressed and I will raise the score to accept (7).

---

> > > > > ### Author Response · Authors · 2024-08-12
> > > > > **Thanks for your comments**
> > > > >
> > > > > Thank you very much for your thoughtful comments and appreciation of our work. We will do our best to improve the final version of our paper based on your valuable suggestions.

---

### Official Review · Reviewer_mUCn · 2024-07-12

**Soundness:** 2
**Presentation:** 2
**Contribution:** 2
**Rating:** 5
**Confidence:** 5

**Summary:**

The paper introduces Deep Degradation Response (DDR), a method to quantify changes in image deep features under varying degradation conditions. DDR facilitates flexible and adaptive degradation through text-driven prompts. It reports to excel in blind image quality assessment and image restoration tasks like deblurring and super-resolution. The paper compares the proposed DDR with the existing techniques across multiple datasets. The authors plan to release their code for public use.

**Strengths:**

1. DDR demonstrates effectiveness across multiple applications, including Blind Image Quality Assessment (BIQA) and image restoration. Its adaptability to different degradation scenarios is a significant advantage.
2. The text-driven approach allows DDR to adjust degradation levels based on specific requirements, making it versatile and applicable to various use cases.

**Weaknesses:**

1. The performance on BIQA is not particularly competitive, and some of the latest IQA metrics, such as LIQE, UNIQUE, and TreS, are not included for comparison. Please refer to the paper “Blind Image Quality Assessment via Vision-Language Correspondence: A Multitask Learning Perspective”.
2. In Equation 8, DDR is calculated using the restored image and its corresponding degradation. According to the results presented in Table 4, the proposed DDR performs better than PSNR+LPIPS. Since LPIPS directly minimizes the feature difference between the restored image and the original image, it should theoretically be more effective for image restoration. The paper does not explain why DDR outperforms the LPIPS loss. Moreover, in Table 6, the pre-trained model also influences the performance of DDR. Does this imply that the gains achieved by DDR might be due to a stronger backbone in comparison to LPIPS?
3. The paper utilizes DDR as an image quality assessment metric but does not provide an explanation for why it could represent the quality of an image.

**Questions:**

Given that the negative and positive values are fixed, T_d)is a constant vector for each degradation. DDR seems to measure the disparity after adding \hat{T_d} to the feature. Does a high DDR indicate that the image distribution is robust to the feature interference caused by T_d?

**Limitations:**

The authors address the limitations and potential societal impact of their work.

---

> ### Author Rebuttal · Authors · 2024-08-07
>
> Thank you for your thoughtful feedback. We address specific points below:
>
> Q1. **Comparison with other BIQA metrics:**
>
> Thank you for the suggestion. The proposed DDR is an Opinion-Unaware Blind Image Quality Assessment (OU-BIQA) metric, which does not require training with human-labeled Mean Opinion Score (MOS) values. In contrast, the LIQE, UNIQUE, and TreS metrics mentioned all require training with MOS values from the IQA dataset. Therefore, we compare DDR with other state-of-the-art OU-BIQA metrics to ensure a fair comparison. Based on your suggestion, we will discuss these works in our revision.
>
> Q2. **Performance comparison between DDR and LPIPS:**
>
> We address this question in the following two parts:
>
> - Q2.1: **Possible reason for DDR outperforms LPIPS**
>
>   We also find it very interesting that DDR outperforms LPIPS. The possible reasons for this may be as follows: 1) DDR leverages the rich multi-modality prior knowledge of CLIP to build joint image-text guidance as a loss for model training; and 2) DDR is a self-supervised learning objective that enhances the model's generalization ability. As a result, the model trained with DDR may perform better on the test set, which could include unseen data distributions.
>
> - Q2.2: **Does the performance gains result from a stronger visual backbone?**
>
>   As shown in Table 2 (in separate PDF file), a stronger backbone in DDR does not always lead to improved performance. For instance, RN50x16 outperforms RN50x64, and ViT-B/32 and ViT-B/16 both outperform ViT-L/14. We anticipate this is because a larger visual model may not necessarily enhance the ability to understand low-level textures. Therefore, we argue that the performance gains achieved by DDR are not due to a stronger vision backbone compared to LPIPS. For the possible reasons please refer to Q2.1.
>
> Q3. **Why DDR can assess image quality?**
>
> Thank you for the thoughtful question. We propose DDR to measure the disparity in image features after introducing degradation in the feature domain. Our findings suggest that images with less degradation (*i.e.*, higher quality) are more sensitive to newly introduced degradation, while images with more degradation (*i.e.*, lower quality) experience less change in their features after further degradation. As shown in Figure 1 and Figure 4 (in the original manuscript), for blurred images, a lower DDR indicates greater blurriness, while a higher DDR corresponds to clearer images. Therefore, we believe DDR effectively quantifies the degree of degradation in images, which enables the assessment of image quality. The extensive experimental results on OU-BIQA presented in Table 3 (in the original manuscript) demonstrate the effectiveness of the proposed DDR as an OU-BIQA model.
>
> Q4. **Does a high DDR indicate that the image distribution is robust to the feature interference caused by $T_d$?**
>
> We agree that there is a correlation between DDR and robustness to feature interference caused by $T_d$. However, a high DDR indicates a significant change in deep image features after introducing degradation in the feature domain, suggesting that the image features are sensitive to the interference caused by $T_d$. In contrast, a low DDR implies that the image is more robust to the interference from $T_d$.

---

> ### Comment · Reviewer_mUCn · 2024-08-12
>
> Thanks for authors' response on the comments. Most of my concerns have been addressed.

---

> > ### Author Response · Authors · 2024-08-12
> > **Thanks for your comments**
> >
> > Thank you very much for your thoughtful comments and appreciation of our work. We will do our best to improve the final version of our paper based on your valuable suggestions.

---

### Official Review · Reviewer_Y3EC · 2024-07-13

**Soundness:** 2
**Presentation:** 2
**Contribution:** 3
**Rating:** 5
**Confidence:** 4

**Summary:**

In this paper, the authors propose a feature descriptor to assess low-level image quality degradations. Based on CLIP, the proposed method first encodes input image and its degraded version to features in CLIP space; the input image is encoded by CLIP image encoder, and the degraded image feature is generated by adding a textual feature of degradation to the image feature. The Deep Degradation Response (DDR) is measured by calculating the distance (seems that cosine is used) of two features. The authors demonstrated the effectiveness of the proposed descriptor with extensive experiments and analysis including SRCC test and applications to image deblurring and super-resolution.

**Strengths:**

1. The paper proposes to exploit CLIP feature to measure image quality under various degradations. Unlike other image quality works focusing on image-based approaches, this paper introduces a novel approach using textual features.
2. Surprisingly, it seems that the proposed method works well without training CLIP with degradation prompts (I need clarification of it in Weakness). It demonstrates that CLIP can be used as a tool for image quality assessment.

**Weaknesses:**

1. Clarity: Some technical details are unclear, which limited the understanding of the proposed method.
- I did not understand the distribution of DDR in Figure 2. Why is the distribution of "adaptive" better than "low" and "high"? What did the authors do specifically for "optimal"?
- The authors mention in L84-85 that there are options in the disparity metric M, but in SRCC evaluation there exist a clear positive or negative direction of correlation. Since Ln and cosine metrics have different meaning for low/high values, more clarification is needed in the description of the method. It seems that the authors used cosine metric in their code.
- It seems that the method simply used pre-trained CLIP features for both images and texts. Is it correct?
- Among degradation types, Color and Content seem ambiguous. Do the authors have a clear definition of these categories?
2. Even though the DDR can score good and bad quality images in terms of (color, noise, blur, ...), it seems difficult to measure the amount of degradation such as noise level. It makes me wonder if the DDR contains useful information to measure degradations, or simply focus on visually pleasing images because positive texts usually correspond to those images? This is a reasonable question as the authors are aware of the fact that the CLIP features may have bias to high-level features rather than low-level degradations.
3. Continuing 2., although the authors framed the paper as image descriptor for degradations, I think the paper is more relevant to non-reference image quality assessment. Therefore, more prior works in this line of research need to be discussed in Sec. 2.
4. Although it is mentioned in the limitation, the proposed method is evaluated on one type of sentence per each degradation. Can the authors justify why they chose these specific words? Due to the nature of texts, there would be similar words that share similar meanings. How robust is the proposed method against the choice of words?

**Questions:**

The paper generally addresses an interesting problem of image quality descriptor using CLIP textual features, and the results look encouraging. However, it lacks the in-depth understanding of the CLIP and proposed method. I would like to hear the authors' answers to the points raised in Weaknesses before adjusting my rating.

**Limitations:**

The authors described two limitations of the proposed method. A potential negative societal impact would be
- The proposed method could be potentially used to discriminate a certain group of photos (e.g. race, gender, etc) by including those words in degradation prompts.

---

> ### Author Rebuttal · Authors · 2024-08-07
>
> We thank the reviewer for the thoughtful response to our paper. We address specific points below:
>
> Q1. **Clarifying of some technical details:**
>
> - **Explanation of Figure 2:** We measure the amount of images with different Degradation Response (DDR) values to represent the distribution of DDR. Specifically, we divided the range of DDR into multiple intervals. For each point on the curve, we measured the number of images whose DDR values fall within the corresponding interval. The horizontal axis in Figure 2 represents the numerical values of DDR, while the vertical axis represents the number of images. By adjusting the levels of handcrafted degradation, DDR demonstrates varying performance on the Opinion-Unaware Blind Image Quality Assessment (OU-BIQA) task. We conducted experiments across a range of degradation levels, selecting the level with the best performance on OU-BIQA as “optimal”. “Low” and “high” represent the lowest and highest degradation levels, respectively. When the degradation level is too low, there is only a subtle difference between the $\mathcal{F}$ and $\mathcal{F}_d$ for all images. In contrast, when the degradation level is too high, most images demonstrate an overly strong response. Using our text-driven “adaptive” strategy, DDR demonstrates a similar value distribution and performance to the manually set “optimal” degradation level. This result shows the effectiveness and flexibility of the proposed method.
>
> - **Clarification of metric M:** We use the cosine distance in our method, which is defined as:
>   $$
>   \mathcal{L}\_{cos}(x, y)=1-\mathcal{S}\_{cos}(x, y),
>   $$
>   where $\mathcal{S}_{cos}(x,y) = \frac{x.y}{||x||||y||}$ represents the cosine metric or cosine similarity between $x$ and $y$. Therefore, the Ln norm and cosine distance adopted in this paper have the same meaning for low/high values. Specifically, a larger Ln norm or cosine distance implies a greater difference, indicating that the image features undergo a more significant change after introducing degradation. Conversely, a smaller Ln norm or cosine distance indicates a smaller discrepancy.
>
> - **Usage of pre-trained CLIP features:** Yes, we directly use the pre-trained CLIP features. We will provide a detailed discussion in response to your Question 2.
> - **Definition of degradation types:** We define the degradation types according to popular papers [r2, r3] (in this response letter). Specifically, “color” degradation refers to unnatural or unpleasant color distortions, such as contrast errors. “Content” degradation refers to issues that result in unclear content, such as down-sampling and JPEG compression artifacts.
>
> Q2. **Whether DDR contains useful degradation information:**
>
> Thank you for your valuable question. We agree with you that CLIP features contain rich high-level information. However, recent works [r1, r2, r3] (in this response letter) have demonstrated that the pre-trained CLIP encoder also possesses a certain ability to understand low-level degradation features. Following these works, we propose to use the pre-trained CLIP encoder to obtain degradation representation in this paper. We acknowledge that fine-tuning CLIP with degradation prompts may lead to better performance. However, a key point of our paper is to reveal the effectiveness of degradation response, and we will explore how to better obtain degradation representations in future work.
>
> [r1] Wang, Jianyi, Kelvin CK Chan, and Chen Change Loy. "Exploring clip for assessing the look and feel of images." Proceedings of the AAAI Conference on Artificial Intelligence. Vol. 37. No. 2. 2023.
>
> [r2] Wu, Haoning, et al. "Towards explainable in-the-wild video quality assessment: a database and a language-prompted approach." Proceedings of the 31st ACM International Conference on Multimedia. 2023.
>
> [r3] Zhang, Weixia, et al. "Blind image quality assessment via vision-language correspondence: A multitask learning perspective." Proceedings of the IEEE/CVF Conference on Computer Vision and Pattern Recognition, 2023.
>
> Q3. **Discussion of more BIQA works:**
>
> Thank you for the clarification. We agree with you that the proposed DDR is highly relevant to non-reference image quality assessment. Specifically, our proposed DDR belongs to Opinion-Unaware Blind Image Quality Assessment (OU-BIQA), which does not require training with human-labeled Mean Opinion Score (MOS) values. Based on your suggestion, we will discuss more relevant prior papers in Section 2.
>
> Q4. **Selection of degradation prompt:**
>
> The proposed DDR is robust to the selection of degradation prompts. We demonstrate this robustness by evaluating DDR with a set of similar words on the OU-BIQA task. As shown in the Table 1 (in separate PDF file), altering positive and negative words results in only minor changes in the SRCC metric for the OU-BIQA task. This observation indicates the robustness of DDR to the choice of positive/negative words. Therefore, we simply select the words that represent each type of degradation with the best performance in our experiments.
>
> Q5. **Potential negative societal impact:**
>
> Thank you for the suggestion. We will discuss this point in our revision.

---

> > ### Author Response · Authors · 2024-08-13
> > **Thanks for your comments**
> >
> > Thank you very much for your valuable comments. We hope our responses have addressed your concerns, and we would be happy to respond to any further queries.
> >
> > Thank you!

---

> > > ### Comment · Reviewer_Y3EC · 2024-08-13
> > >
> > > Thank the authors for addressing the raised concerns. I have read all reviews and the authors' responses. All of my concerns are resolved and I think the paper has a valid contribution to be published at the conference. I will increase my rating to borderline accept.

---

> > > > ### Author Response · Authors · 2024-08-13
> > > > **Thanks for your comments**
> > > >
> > > > Thank you very much for your thoughtful comments and appreciation of our work. We will do our best to improve the final version of our paper based on your valuable suggestions.
> > > >
> > > > Thank you!

---

### Author Rebuttal · Authors · 2024-08-07

We thank the reviewers for their insightful feedback, which has significantly improved our paper. We are delighted that they appreciate the following: “*This paper introduces a novel approach using textual features, …, the results look encouraging.*” (**Reviewer Y3EC**) “*The adaptability to different degradation scenarios is a significant advantage.*” (**Reviewer mUCn**) “*The experiment is sufficient and detailed.*” (**Reviewer tuJD**)

The main concerns of the paper include an in-depth analysis of the proposed method (Reviewer Y3EC, Reviewer mUCn), more comparison with existing work (Reviewer mUCn, Reviewer tuJD), technical clarification and more experimental details (Reviewer Y3EC, Reviewer tuJD). Our responses to these questions and suggestions can be summarized as follows:

- **In-depth analysis of the proposed method:** We discussed the usage of the pre-trained CLIP model and the selection of the degradation prompt in response to Reviewer Y3EC. Additionally, we analyzed the performance comparison between DDR and LPIPS in response to Reviewer mUCn.
- **More comparison with existing work:** We elaborated on the differences between the proposed DDR and existing methods in response to Reviewer mUCn and Reviewer tuJD.
- **Technical clarification and more experimental details:** We offered technical clarifications following the suggestions of Reviewer Y3EC and detailed experimental settings regarding training resolution and learning rate in response to Reviewer tuJD.
- **Additional experiments:** We conducted additional experiments on prompt settings, the selection of vision backbones, and testing on other SISR models in separate responses to each reviewer, respectively. Tables containing all additional experiment result are compiled into a separate PDF file. Please download and refer to this PDF file as needed.

---

### Decision · Program_Chairs · 2024-09-25

**Decision:**

Accept (poster)

**Comment:**

The final scores for this work are two borderline accept and one accept. All reviewers showed a generally positive attitude towards the work, with no significant disagreements. After the authors provided their rebuttal, all reviewers clearly indicated that their previous concerns had been addressed and chose to upgrade their ratings. In summary, I am inclined to accept this work.